# A Sanity Check for AI-generated Image Detection

**Shilin Yan**[1*], **Ouxiang Li**[2*], **Jiayin Cai**[1*], **Yanbin Hao**[2], **Xiaolong Jiang**[1], **Yao Hu**[1], **Weidi Xie**[3†]

[1]Xiaohongshu Inc. [2]University of Science and Technology of China [3]Shanghai Jiao Tong University
`tattoo.ysl@gmail.com`   `weidi@sjtu.edu.cn`

Project Page: https://shilinyan99.github.io/AIDE/

## Abstract

With the rapid development of generative models, discerning AI-generated content has evoked increasing attention from both industry and academia. In this paper, we conduct a sanity check on **whether the task of AI-generated image detection has been solved**. To start with, we present `Chameleon` dataset, consisting of AI-generated images that are genuinely challenging for human perception. To quantify the generalization of existing methods, we evaluate 9 off-the-shelf AI-generated image detectors on `Chameleon` dataset. Upon analysis, almost all models misclassify AI-generated images as real ones. Later, we propose **AIDE** (**AI**-generated **I**mage **DE**tector with Hybrid Features), which leverages multiple experts to simultaneously extract visual artifacts and noise patterns. Specifically, to capture the high-level semantics, we utilize CLIP to compute the visual embedding. This effectively enables the model to discern AI-generated images based on semantics and contextual information. Secondly, we select the highest and lowest frequency patches in the image, and compute the low-level patchwise features, aiming to detect AI-generated images by low-level artifacts, for example, noise patterns, anti-aliasing effects. While evaluating on existing benchmarks, for example, AIGCDetectBenchmark and GenImage, AIDE achieves **+3.5**% and **+4.6**% improvements to state-of-the-art methods, and on our proposed challenging `Chameleon` benchmarks, it also achieves promising results, despite the problem of detecting AI-generated images remains far from being solved.

## 1 Introduction

Recently, the vision community has witnessed remarkable advancements in generative models. These methods, ranging from generative adversarial networks (GANs) (Goodfellow et al., 2014; Zhu et al., 2017; Brock et al., 2019; Karras et al., 2019) to diffusion models (DMs) (Ho et al., 2020; Nichol & Dhariwal, 2021; Rombach et al., 2022; Song et al., 2021; Liu et al., 2022b; Lu et al., 2022; Hertz et al., 2023; Nichol et al., 2022) have demonstrated unprecedented capabilities in synthesizing high-quality images that closely resemble real-world scenes. On the positive side, such generative models have enabled various valuable tools for artists and designers, democratizing access to advanced graphic design capabilities. However, it also raises concerns about the authenticity of visual content, posing significant challenges for image forensics (Ferreira et al., 2020), misinformation combating (Xu et al., 2023a), and copyright protection (Ren et al., 2024). In this paper, we consider the problem of distinguishing between images generated by AI models and those originating from real-world sources.

In the literature, although there are numerous AI-generated image detectors (Wang et al., 2020; Frank et al., 2020; Ojha et al., 2023; Wang et al., 2023a; Zhong et al., 2023; Ricker et al., 2024) and benchmarks (Wang et al., 2020; 2023a; Zhu et al., 2024; Hong & Zhang, 2024), the prevailing problem formulation typically involves training models on images generated solely by GANs

---

* Equal contribution.    † Corresponding author.

(*e.g.*, ProGAN (Karras et al., 2018)) and evaluating their performance on datasets including images from various generative models, including GANs and DMs. However, such formulation poses two fundamental issues in practice: (i) evaluation benchmarks are simple, as they often feature test sets composed of random images from generative models, rather than images that present genuine challenges for human perception; (ii) confining models to train exclusively on images from certain types of generative models (GANs or DMs) imposes an unrealistic constraint, hindering the model's ability to learn from the diverse properties exhibited by more advanced generative models.

To address the aforementioned issues, we propose two pivotal strategies. *Firstly*, we introduce a novel testset for AI-generated image detection, named **Chameleon**, manually annotated to include images that genuinely challenge human perception. This dataset has three key features: (i) deceptively real: all AI-generated images in the dataset have passed a human perception "Turing Test", *i.e.*, human annotators have misclassified them as real images. (ii) diverse categories: comprising images of human, animal, object, and scene categories, the dataset depicts real-world scenarios across various contexts. (iii) high resolution: with most images having resolutions exceeding 720P and going up to 4K, all images in the dataset exhibit exceptional clarity. Consequently, this test set offers a more realistic evaluation of model performance. After evaluating 9 off-the-shelf AI-generated image detectors on **Chameleon**, unfortunately, all detectors suffer from significant performance drops, mis-classifying the AI-generated images as real ones. *Secondly*, we reformulate the AI-generated image detection problem setup, which enables models to train across a broader spectrum of generative models, enhancing their adaptability and robustness in real-world scenarios.

Based on the above observation, it is clear that detecting AI-generated images remains challenging, and far from being solved. Therefore, a fundamental question arises: what distinguishes AI-generated images from real ones? Intuitively, such cues may appear from various aspects, including low-level textures or pixel statistics (*e.g.*, *the presence of white noise during image capturing*), and high-level semantics (*e.g.*, *penguins are unlikely to be appearing on the grassland in Africa*), geometry principle (*e.g.*, perspective), physics (*e.g.*, lighting condition). To reflect such intuition, we propose a simple AI-generated image detector, termed as **AIDE** (**AI**-generated **I**mage **DE**tector with Hybrid Features). Specifically, **AIDE** incorporates a DCT (Ahmed et al., 1974) scoring module to capture low-level pixel statistics by extracting both high and low-frequency patches from the image, which are then processed through SRM (Spatial Rich Model) filters (Fridrich & Kodovsky, 2012) to characterize the noise pattern. Additionally, to capture global semantics, we utilize the pre-trained OpenCLIP (Ilharco et al., 2021) to encode the entire image. The features from various levels are fused in the channel dimension for the final prediction. To evaluate the effectiveness of our model, we conduct extensive experiments on two popular benchmarks, including AIGCDetectBenchmark (Wang et al., 2020) and GenImage (Zhu et al., 2024), for AI-generated image detection. On AIGCDetectBenchmark and GenImage benchmarks, AIDE surpasses state-of-the-art (SOTA) methods by **+3.5%** and **+4.6%** in accuracy scores, respectively. Moreover, AIDE also achieves competitive performance on our **Chameleon** benchmark.

Overall, our contributions are summarized as follows: (i) we present the **Chameleon** dataset, a meticulously curated test set designed to challenge human perception by including images that deceptively resemble real-world scenes. With thorough evaluation of 9 different off-the-shelf detectors, this dataset exposes the limitations of existing approaches; (ii) we present a simple mixture-of-expert model, termed AIDE, that enables to discern AI-generated images based on both low-level pixel statistics and high-level semantics; (iii) experimentally, our model achieves state-of-the-art results on public benchmarks for AIGCDetectBenchmark (Wang et al., 2020) and GenImage (Zhu et al., 2024). While on **Chameleon**, it acts as a competitive baseline on a realistic evaluation benchmark, to foster future research in this community.

## 2 RELATED WORKS

**AI-generated Image Detection.** The demand for detecting AI-generated images has long been present. Early studies primarily focus on spatial domain cues, such as color (McCloskey & Albright, 2018), saturation (McCloskey & Albright, 2019), co-occurrence (Nataraj et al., 2019), and reflections (O'brien & Farid, 2012). However, these methods often suffer from limited generalization capabilities as generators progress. To address this limitation, CNNSpot (Wang et al., 2020) discovers that an image classifier trained exclusively on ProGAN (Karras et al., 2018) generator could gener-

alize effectively to other unseen GAN architectures, with careful pre- and post-processing and data augmentation. FreDect (Frank et al., 2020) observes significant artifacts in the frequency domain of GAN-generated images, attributed to the upsampling operation in GAN architectures. More recent approaches have explored novel perspectives for superior generalization ability. UnivFD (Ojha et al., 2023) proposes to train a universal liner classifier with pretrained CLIP-ViT (Dosovitskiy et al., 2020; Radford et al., 2021) features. DIRE (Wang et al., 2023a) introduces DIRE features, which computes the difference between images and their reconstructions from pretrained ADM (Dhariwal & Nichol, 2021), to train a deep classifier. PatchCraft (Zhong et al., 2023) compares rich-texture and poor-texture patches from images, extracting the inter-pixel correlation discrepancy as a universal fingerprint, which is reported to achieve the state-of-the-art (SOTA) generalization performance. AEROBLADE (Ricker et al., 2024) proposes a training-free detection method for latent diffusion models using autoencoder reconstruction errors. FatFormer (Liu et al., 2024a) introduces a forgery-aware adapter to discern and integrate local forgery traces based on CLIP. CLIPMoLE (Liu et al., 2024b) adapts a combination of shared and separate LoRAs within an MoE-based structure in deeper ViT blocks. SSP (Chen et al., 2024) employs the simplest patches for detection. However, these methods only discriminate real or fake images from a single perspective, often failing to generalize across images from different generators.

**AI-generated Image Datasets.** To facilitate AI-generated image detection, many datasets containing both real and fake images have been organized for training and evaluation. Early dataset from CNNSpot (Wang et al., 2020) collects fake images from GAN series generators (Goodfellow et al., 2014; Zhu et al., 2017; Brock et al., 2019; Karras et al., 2019). Particularly, this dataset generates fake images exclusively using ProGAN (Karras et al., 2018) as training data and evaluates the generalization ability on a set of GAN-based testing data. However, with recent emergence of more advanced generators, such as diffusion model (DM) (Ho et al., 2020) and its variants (Dhariwal & Nichol, 2021; Nichol & Dhariwal, 2021; Rombach et al., 2022; Song et al., 2021; Liu et al., 2022b; Lu et al., 2022; Hertz et al., 2023; Nichol et al., 2022), their realistic generations make visual differences between real and fake images progressively harder to detect. Subsequently, more datasets including DM-generated images have been proposed one after another, including DE-FAKE (Xu et al., 2023b), CiFAKE (Bird & Lotfi, 2024), DiffusionDB (Wang et al., 2023b), ArtiFact (Rahman et al., 2023). One representative dataset is GenImage (Zhu et al., 2024), which comprises ImageNet's 1,000 classes generated using 8 SOTA generators in both academia (*e.g.,* Stable Diffusion (Sta, 2022)) and industry (*e.g.,* Midjourney (mid)). More recently, Hong *et al.* introduce a more comprehensive dataset, WildFake (Hong & Zhang, 2024), which includes AI-generated images sourced from multiple generators, architectures, weights, and versions. However, existing benchmarks only evaluate AI-generated images using current foundational models with simple prompts and few modifications, whereas deceptively real images from online communities usually necessitate hundreds to thousands of manual parameter adjustments.

# 3 CHAMELEON DATASET

## 3.1 PROBLEM FORMULATION

In this paper, our goal is to train a model that can distinguish the AI-generated images from the ones captured by the camera, *i.e.,* $y = \Phi_{\text{model}}(\mathbf{I}; \Theta) \in \{0, 1\}$, where $\mathbf{I} \in \mathbb{R}^{H \times W \times 3}$ denotes an input RGB image, $\Theta$ refers to the learnable parameters. For training and testing, we consider the following two settings:

**Train-Test Setting-I.** In the literature, existing works on detecting AI-generated images (Wang et al., 2020; Frank et al., 2020; Ojha et al., 2023; Wang et al., 2023a; Zhong et al., 2023) have exclusively considered the scenario of training on images from single generative model, for example, ProGAN (Karras et al., 2018), or Stable Diffusion (Sta, 2022), and then evaluated on images from various generative models. That is,

$$\mathcal{G}_{\text{train}} = \mathcal{G}_{\text{GAN}} \vee \mathcal{G}_{\text{DM}}, \quad \mathcal{G}_{\text{test}} = \{\mathcal{G}_{\text{ProGAN}}, \mathcal{G}_{\text{CycleGAN}}, ..., \mathcal{G}_{\text{SD}}, \mathcal{G}_{\text{Midjourney}}\} \quad (1)$$

Generally speaking, such problem formulation poses two critical issues: (i) evaluation benchmarks are simple, as these randomly sampled images from generative models, can be far from being photorealistic, as shown in Figure 1; (ii) confining models to train exclusively on GAN-generated images imposes an unrealistic constraint, hindering the model's ability to learn from the diverse properties exhibited by more advanced generative models.

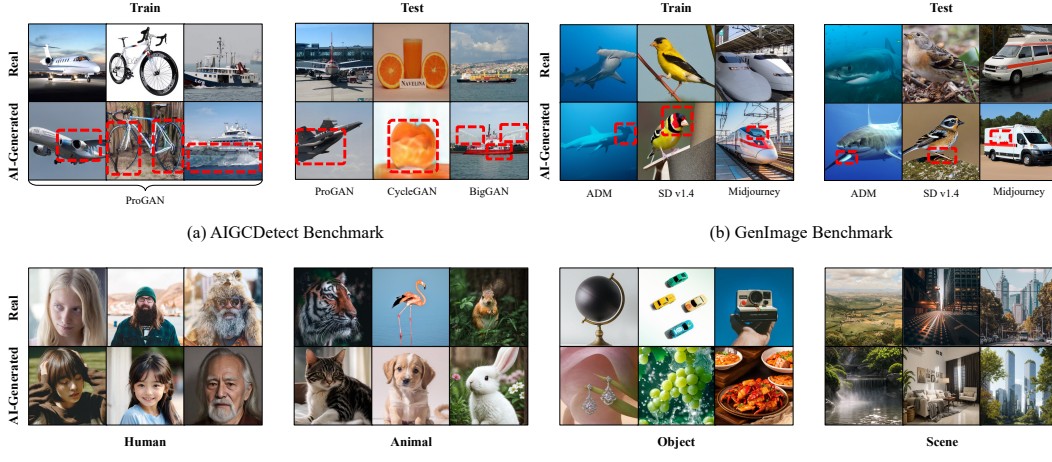

Figure 1: **Comparison of `Chameleon` with existing benchmarks.** We visualize two contemporary AI-generated image benchmarks, namely (a) AIGCDetect Benchmark Wang et al. (2020) and (b) GenImage Benchmark Zhu et al. (2024), where all images are generated from publicly available generators, such as Pro-GAN (GAN-based), SD v1.4 (DM-based) and Midjourney (commercial API). These images are generated by unconditional situations or conditioned on simple prompts (e.g., *photo of a plane*) without delicate manual adjustments, thereby inclined to generate obvious artifacts in consistency and semantics (marked with red boxes). In contrast, our `Chameleon` dataset in (c) aims to simulate real-world scenarios by collecting diverse images from online websites, where these online images are carefully adjusted by photographers and AI artists.

Table 1: **Statistics of the Chameleon testset,** including over 11k high-fidelity AI-generated images from art; civ; lib, as well as a similar scale of real-world photographs from uns.

|  | Scene | Object | Animal | Human | Total |
|---|---|---|---|---|---|
| **Real Images** | 3,574 | 3,578 | 3,998 | 3,713 | 14,863 |
| **Fake Images** | 2,976 | 2,016 | 313 | 5,865 | 11,170 |

**Train-Test Setting-II.** Herein, we propose an alternative problem formulation, where the models are allowed to train on images generated from a wide spectrum of generative models, and then tested on images that are genuinely challenging for human perception.

$$\mathcal{G}_{\text{train}} = \{\mathcal{G}_{\text{GAN}}, \mathcal{G}_{\text{DM}}\}, \quad \mathcal{G}_{\text{test}} = \{\mathcal{D}_{\text{Chameleon}}\} \tag{2}$$

$\mathcal{D}_{\text{Chameleon}}$ refers to our proposed benchmark, as detailed below. We believe this setting resembles a more practical scenario for future model development in this community.

## 3.2 CHAMELEON DATASET

The primary objective of the `Chameleon` dataset is to evaluate the generalization and robustness of existing AI-generated image detectors, for a sanity check on the progress of AI-generated image detection. In this section, we outline the progression of the proposed dataset, including: (i) collection, (ii) curation, (iii) annotation. The statistical results of our dataset are illustrated in Table 1 and we compare our dataset with existing benchmarks in Fig. 1.

### 3.2.1 DATASET COLLECTION

To simulate the real-world scenarios in detecting AI-generated images, we structure `Chameleon` dataset based on three main principles: (i) images must be deceptively real, and (ii) they should cover a diverse range of categories, and (iii) they should also have very high image quality. Importantly, each image must have (a) Creative Commons (CC BY 4.0) license, or (b) explicit permissions obtained from the owners to use in our research. Herein, we present the details of image collection.

**Fake Image Collection:** To collect images that are deceptively real, and cover sufficiently diverse categories, we source user-created AI-generated images from popular AI-painting communities, *i.e.,* ArtStation (art), Civitai (civ), and Liblib (lib), many of which utilize commercial APIs (*e.g.,* Midjourney (mid) and DALLE-3 (Ramesh et al., 2022)) or various LoRA modules (Hu et al., 2022) with

Table 2: **Comparison of AI-generated image detection testset.** Our `Chameleon` dataset is the first to encompass real-life scenarios for evaluation. Compared to AIGCDetBenchmark (Zhong et al., 2023), `Chameleon` offers greater magnitude and superior quality, rendering it more realistic in evaluation. IN represents ImageNet.

| | ProGAN | StyleGAN | BigGAN | CycleGAN | StarGAN | GauGAN | StyleGAN2 | WFIR | ADM | Glide | Midjourney | SD v1.4 | SD v1.5 | VQDM | Wukong | DALLE2 | Chameleon |
|---|---|---|---|---|---|---|---|---|---|---|---|---|---|---|---|---|---|
| **Magnitude** | 8.0$k$ | 12.0$k$ | 4.0$k$ | 2.6$k$ | 4.0$k$ | 10.0$k$ | 15.9k | 2.0$k$ | 12.0$k$ | 12.0$k$ | 12.0$k$ | 12.0$k$ | 16.0$k$ | 12.0$k$ | 12.0$k$ | 2.0$k$ | **26.0$k$** |
| **Resolution** | 256 | 256 | 256 | 256 | 256 | 256 | 256 | 1024 | 256 | 256 | 1024 | 512 | 512 | 256 | 512 | 256 | **720P-4K** |
| **Variety** | LSUN | LUSN | IN | IN | CelebA | COCO | LSUN | FFHQ | IN | IN | IN | IN | IN | IN | IN | IN | **Real-life** |

Stable Diffusion (SD) (Sta, 2022) that fine-tuned on their in-house data. Specifically, we initiate the process by utilizing GPT-4 (cha, 2022) to generate diverse query words to retrieve AI-generated images. Throughout the collection process, we enforce stringent NSFW (Not Safe For Work) restrictions. Ultimately, our collection comprises over 150K AI-generated images.

**Real Image Collection:** To ensure that real and AI-generated images fall into the same distribution, we employ identical query words to retrieve real images, mirroring the approach used for gathering AI-generated images. Eventually, we collect over 20K images from platforms like Unsplash (uns), which is an online community providing high-quality, free-to-use images contributed by photographers worldwide.

### 3.2.2 DATASET CURATION

To ensure the collection of high-quality images, we implement a comprehensive pipeline for image cleaning: (i) we discard images with resolution lower than $448 \times 448$, as higher-resolution images generally provide better assessments of the robustness of existing models; (ii) due to the potential presence of violent and inappropriate content, we utilize SD's safety checker model (saf, 2022) to filter out NSFW images; (iii) to avoid duplicated images, we compare their hash values to filter out duplicated images. In addition to this general cleaning pipeline, we introduce CLIP (Radford et al., 2021) to further filter out images with low image-text similarity. Specifically, for AI-generated images, the online website provides prompts used to generate these images, and we calculate similarity using their corresponding prompts. For real images, we used the mean of the 80 prompt templates (*e.g., a photo of* {*category*} and *a photo of the* {*category*}) evaluated in CLIP's ImageNet zero-shot as the text embedding.

### 3.2.3 DATASET ANNOTATION

At this stage, we establish an annotation platform and recruit 20 human workers to manually label each of the AI-generated images for their category and realism. For categorization, annotators are instructed to assign each image to one of four major categories: human, animal, object, and scene. Regarding realism assessment, workers are tasked with labeling the images as either **Real** or **AI-generated**, based on the criterion of **whether this image could be taken with a camera**. Note that, the annotators are not informed whether the images are generated by AI algorithms beforehand. Each image was assessed independently by two annotators, and those have been misclassified as real by both annotators can thus be considered to pass the "perception turing test" and labeled as *highly realistic*. Subsequently, we retain only those images judged as *highly realistic*. Similarly, for real images, we follow the same procedure, retaining only those belonging to the four predefined categories, as we have done for AI-generated images.

### 3.2.4 DATASET COMPARISON

Our objective is to construct a sophisticated and exhaustive test set that serves as a valuable extension to the current evaluation benchmarks for AI-generated image detection. In Table 2, we conduct a comparative analysis between our `Chameleon` dataset and existing test sets. Our dataset is characterized by three pivotal features: (i) **Magnitude.** Comprised of approximately 26,000 test images, the `Chameleon` dataset represents the most extensive collection available, surpassing any existing test set and enhancing its robustness. (ii) **Variety.** Our dataset incorporates images from a vast array of real-world scenarios, surpassing the limited categorical focus of the other datasets. (iii) **Resolution.** With resolutions spanning from 720P to 4K, With image resolutions ranging from 720P to 4K, artifacts demand more nuanced analysis, thus presenting additional challenges to the model due to

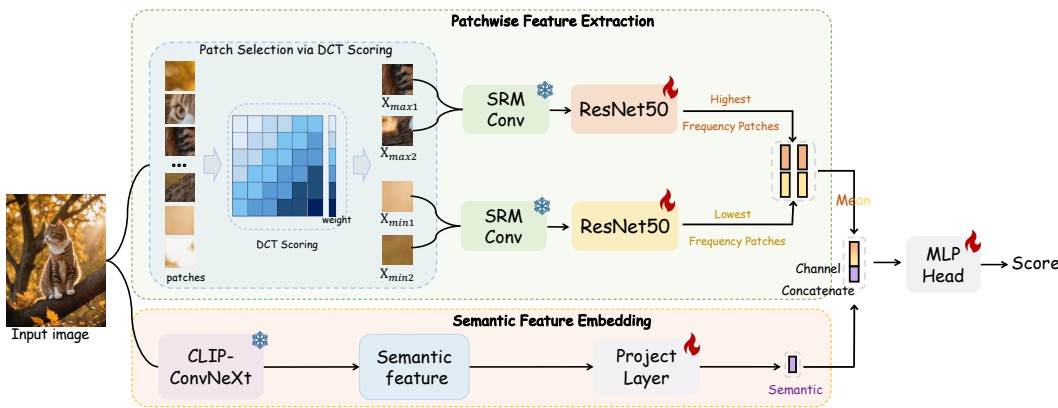

Figure 2: **Overview of AIDE.** It consists of a Patchwise Feature Extraction (PFE) module and a Semantic Feature Embedding (SFE) module in a mixture of experts manner. In PFE module, the DCT Scoring module first calculates the DCT coefficients for each smashed patch and then performs a weighted sum of these coefficients (weights gradually increase as the color goes from light to dark).

the need for fine-grained discernment. In summary, our dataset offers a more demanding and pragmatically relevant benchmark for the advancement of AI-generated image detection methodologies.

## 4 METHODOLOGY

In this section, we present AIDE (**AI**-generated **I**mage **DE**tector with Hybrid Features), consisting of a module to compute low-level statistics for texture or smooth patches, a high-level semantic embedding module, and a discriminator to classify the image as being generated or photographed. The overview of our AIDE model is illustrated in Fig. 2.

### 4.1 PATCHWISE FEATURE EXTRACTION

Here, our design leverages the disparities in low-level patch statistics between AI-generated images and real-world scenes. Models like generative adversarial networks or diffusion models often yield images with certain artifacts, such as excessive smoothness or anti-aliasing effects. To capture such discrepancy, we adopt a Discrete Cosine Transform (DCT) score module to identify patches with the highest and lowest frequency. By focusing on these extreme patches, we aim to highlight the distinctive characteristics of AI-generated images, thus facilitating the discriminative power of our detection system.

**Patch Selection via DCT Scoring.** For an RGB image, we first divide this image into multiple patches with a fixed window size, $\mathbf{I} = \{x_1, x_2, \ldots, x_n\}$, $x_i \in \mathbb{R}^{N \times N \times 3}$. In our case, the patch size is set to be $N = 32$ pixels. We apply the discrete cosine transform to each of the image patches, obtaining the corresponding results in the frequency domain, $\mathcal{X}_f = \{x_1^{\text{dct}}, x_2^{\text{dct}} \ldots, x_n^{\text{dct}}\}$, $x_i^{\text{dct}} \in \mathbb{R}^{N \times N \times 3}$.

To acquire the highest and lowest image patches, we use the complexity of the frequency components as an indicator. From this, we design a simple yet effective scoring mechanism, using $K$ different band-pass filters:

$$F_{k,ij} = \begin{cases} 1, & \text{if } \frac{2N}{K} \cdot k \leq i + j < \frac{2N}{K} \cdot (k+1) \\ 0, & \text{otherwise} \end{cases} \tag{3}$$

where $F_{k,ij}$ is the weight at the $(i, j)$ position of the $k$-th filter. Next, for the $m$-th patch ($x_m^{\text{dct}} \in \mathbb{R}^{N \times N \times 3}$), we apply the filters $F_{k,ij} \in \mathbb{R}^{N \times N \times 3}$ to multiply the logarithm of the absolute DCT coefficients ($x_m^{\text{dct}} \in \mathbb{R}^{N \times N \times 3}$) and sum up all the positions to obtain the grade of the patch $G^m$. We formulated it as

$$G^m = \sum_{k=0}^{K-1} 2^k \times \sum_{c=0}^{2} \sum_{i=0}^{N-1} \sum_{j=0}^{N-1} F_{k,ij} \cdot \log(|x_m^{\text{dct}}| + 1) \tag{4}$$

where $c$ is the number of patch channels. In this way, we acquire the grades of all patches $G = \{G^1, G^2, ..., G^n\}$. We then sort them to identify the highest and lowest frequency patches.

Through the scoring module, we can obtain the top $k$ patches $X_{\max} = \{X_{\max_1}, X_{\max_2}, ..., X_{\max_k}\}$ with the highest frequency and the top $k$ patches $X_{\min} = \{X_{\min_1}, X_{\min_2}, ..., X_{\min_k}\}$ with the lowest frequency, where $X_{\max_i} \in \mathbb{R}^{N \times N \times 3}$, $X_{\min_i} \in \mathbb{R}^{N \times N \times 3}$.

**Patchwise Feature Encoder.** These patches are resized to size of $256 \times 256$ pixels, and input into the SRM (Fridrich & Kodovsky, 2012) to extract their noise pattern. Lastly, these features are further input into two ResNet-50 (He et al., 2016) networks ($f_1(\cdot)$ and $f_2(\cdot)$) to obtain the final feature map $F_{\max} = \{f_1(X_{\max_1}), f_1(X_{\max_2}), ..., f_1(X_{\max_k})\}$, $F_{\min} = \{f_2(X_{\min_1}), f_2(X_{\min_2}), ..., f_2(X_{\min_k})\}$. We acquire the highest and lowest frequency embedding on the mean-pooled feature:

$$F_{\max} = \text{Mean}(\text{AveragePool}(F_{\max})), \quad F_{\min} = \text{Mean}(\text{AveragePool}(F_{\min})) \tag{5}$$

### 4.2 SEMANTIC FEATURE EMBEDDING

To capture the rich semantic features within images, such as object co-occurrence and contextual relationships, we compute the visual embedding for input image with an off-the-shelf visual-language foundation model. Specifically, we adopt the ConvNeXt-based OpenCLIP model (Ilharco et al., 2021) to get the final feature map ($v \in \mathbb{R}^{h \times w \times c}$). To capture the global contexts, we append a linear projection layer followed by mean spatial pooling:

$$F_s = \text{avgpool}(g(v)). \tag{6}$$

### 4.3 DISCRIMINATOR

To distinguish between AI-generated images and real images, we utilize a mixture-expert model for the final discrimination. At low-level, we take the average of the highest frequency featured:

$$F_{\text{mean}} = \text{avgpool}(F_{\max}, F_{\min}). \tag{7}$$

Then, we channel-wisely concatenate the representations between it and high-level embedding $F_s$. At last, the features are encoded into MLP to acquire the score,

$$y = f([\text{avgpool}(F_{\text{mean}}; F_s]) \tag{8}$$

where $f(\cdot)$ denotes the MLP consisting of a linear layer, GELU (Hendrycks & Gimpel, 2016) and classifier, $[;]$ refers to the operation of channel-wise concatenation.

## 5 EXPERIMENTS

### 5.1 EXPERIMENTAL DETAILS

**Baseline Detectors.** We evaluate 9 off-the-shelf detectors including CNNSpot (Wang et al., 2020), FreDect (Frank et al., 2020), Fusing (Ju et al., 2022), LNP (Liu et al., 2022a), LGrad (Tan et al., 2023), UnivFD (Ojha et al., 2023), DIRE (Wang et al., 2023a), PatchCraft (Zhong et al., 2023) and NPR (Tan et al., 2024) for comparison.

**Datasets.** To comprehensively evaluate the generalization ability of existing approaches, we conduct experiments across two kinds of settings: **Setting-I** and **Setting-II**, which are summarized in Sec. 3.1. For the **Setting-I** setting, we evaluate the detectors on two general and comprehensive benchmarks of AIGCDetectBenchmark ($\mathcal{B}_1$) (Zhong et al., 2023) and GenImage (Zhu et al., 2024) ($\mathcal{B}_2$). For the **Setting-II** setting, we evaluate the detectors on our **Chameleon** ($\mathcal{B}_3$) benchmark. More details can be found in Appendix.

**Implementation Details.** AIDE includes two key modules: Patchwise Feature Extraction (PFE) and Semantic Feature Embedding (SFE). For PFE channel, we first patchify each image into patches and the patch size is set to be $N = 32$ pixels. Then these patches are sorted using our DCT Scoring module with $K = 6$ different band-pass filters in the frequency domain. Subsequently, we select two highest-frequency and two lowest-frequency patches using the calculated DCT scores. These selected patches are then resized to $256 \times 256$ and extracted their noise pattern using SRM (Fridrich & Kodovsky, 2012). For SFE channel, we use the pre-trained OpenCLIP (Ilharco et al., 2021) to

Table 3: **Comparison on the AIGCDetectBenchmark** (Zhong et al., 2023). Accuracy (%) of different detectors (rows) in detecting real and fake images from different generators (columns). DIRE-D indicates this result comes from DIRE detector trained over fake images generated by ADM following its official setup (Wang et al., 2023a). DIRE-G indicates this baseline is trained on the same ProGAN training data as others. The best result and the second-best result are marked in **bold** and underline, respectively.

| Method | ProGAN | StyleGAN | BigGAN | CycleGAN | StarGAN | GauGAN | StyleGAN2 | WFIR | ADM | Glide | Midjourney | SD v1.4 | SD v1.5 | VQDM | Wukong | DALLE2 | Mean |
|---|---|---|---|---|---|---|---|---|---|---|---|---|---|---|---|---|---|
| CNNSpot | **100.00** | 90.17 | 71.17 | 87.62 | 94.60 | 81.42 | 86.91 | 91.65 | 60.39 | 58.07 | 51.39 | 50.57 | 50.53 | 56.46 | 51.03 | 50.45 | 70.78 |
| FreDect | 99.36 | 78.02 | 81.97 | 78.77 | 94.62 | 80.57 | 66.19 | 50.75 | 63.42 | 54.13 | 45.87 | 38.79 | 39.21 | 77.80 | 40.30 | 34.70 | 64.03 |
| Fusing | **100.00** | 85.20 | 77.40 | 87.00 | 97.00 | 77.00 | 83.30 | 66.80 | 49.00 | 57.20 | 52.20 | 51.00 | 51.40 | 55.10 | 51.70 | 52.80 | 68.38 |
| LNP | 99.67 | 91.75 | 77.75 | 84.10 | 99.92 | 75.39 | 94.64 | 70.85 | 84.73 | 80.52 | 65.55 | 85.55 | 85.67 | 74.46 | 82.06 | 88.75 | 83.84 |
| LGrad | 99.83 | 91.08 | 85.62 | 86.94 | 99.27 | 78.46 | 85.32 | 55.70 | 67.15 | 66.11 | 65.35 | 63.02 | 63.67 | 72.99 | 59.55 | 65.45 | 75.34 |
| UnivFD | 99.81 | 84.93 | 95.08 | 98.33 | 95.75 | 99.47 | 74.96 | 86.90 | 66.87 | 62.46 | 56.13 | 63.66 | 63.49 | 85.31 | 70.93 | 50.75 | 78.43 |
| DIRE-G | 95.19 | 83.03 | 70.12 | 74.19 | 95.47 | 67.79 | 75.31 | 58.05 | 75.78 | 71.75 | 58.01 | 49.74 | 49.83 | 53.68 | 54.46 | 66.48 | 68.68 |
| DIRE-D | 52.75 | 51.31 | 49.70 | 49.58 | 46.72 | 51.23 | 51.72 | 53.30 | **98.25** | 92.42 | 89.45 | 91.24 | 91.63 | 91.90 | 90.90 | 92.45 | 71.53 |
| PatchCraft | **100.00** | 92.77 | **95.80** | 70.17 | **99.97** | 71.58 | 89.55 | 85.80 | 82.17 | 83.79 | 90.12 | **95.38** | **95.30** | 88.91 | 91.07 | **96.60** | 89.31 |
| NPR | 99.79 | 97.70 | 84.35 | 96.10 | 99.35 | 82.50 | 98.38 | 65.80 | 69.69 | 78.36 | 77.85 | 78.63 | 78.89 | 78.13 | 76.11 | 64.90 | 82.91 |
| *AIDE* | 99.99 | **99.64** | 83.95 | 98.48 | 99.91 | 73.25 | 98.00 | **94.20** | 93.43 | **95.09** | 77.20 | 93.00 | 92.85 | 95.16 | 93.55 | **96.60** | **92.77** |

Table 4: **Comparison on the GenImage** (Zhu et al., 2024). Accuracy (%) of different detectors (rows) in detecting real and fake images from different generators (columns). These methods are trained on real images from ImageNet and fake images generated by SD v1.4 and evaluated over eight generators. The best result and the second-best result are marked in **bold** and underline, respectively.

| Method | Midjourney | SD v1.4 | SD v1.5 | ADM | GLIDE | Wukong | VQDM | BigGAN | Mean |
|---|---|---|---|---|---|---|---|---|---|
| ResNet-50 (He et al., 2016) | 54.90 | **99.90** | 99.70 | 53.50 | 61.90 | 98.20 | 56.60 | 52.00 | 72.09 |
| DeiT-S (Touvron et al., 2021) | 55.60 | **99.90** | 99.80 | 49.80 | 58.10 | 98.90 | 56.90 | 53.50 | 71.56 |
| Swin-T (Liu et al., 2021) | 62.10 | **99.90** | 99.80 | 49.80 | 67.60 | 99.10 | 62.30 | 57.60 | 74.78 |
| CNNSpot (Wang et al., 2020) | 52.80 | 96.30 | 95.90 | 50.10 | 39.80 | 78.60 | 53.40 | 46.80 | 64.21 |
| Spec (Zhang et al., 2019) | 52.00 | 99.40 | 99.20 | 49.70 | 49.80 | 94.80 | 55.60 | 49.80 | 68.79 |
| F3Net (Qian et al., 2020) | 50.10 | **99.90** | **99.90** | 49.90 | 50.00 | **99.90** | 49.90 | 49.90 | 68.69 |
| GramNet (Liu et al., 2020) | 54.20 | 99.20 | 99.10 | 50.30 | 54.60 | 98.90 | 50.80 | 51.70 | 69.85 |
| DIRE (Wang et al., 2023a) | 60.20 | **99.90** | 99.80 | 50.90 | 55.00 | 99.20 | 50.10 | 50.20 | 70.66 |
| UnivFD (Ojha et al., 2023) | 73.20 | 84.20 | 84.00 | 55.20 | 76.90 | 75.60 | 56.90 | **80.30** | 73.29 |
| GenDet (Zhu et al., 2023) | **89.60** | 96.10 | 96.10 | 58.00 | 78.40 | 92.80 | 66.50 | 75.00 | 81.56 |
| PatchCraft (Zhong et al., 2023) | 79.00 | 89.50 | 89.30 | 77.30 | 78.40 | 89.30 | **83.70** | 72.40 | 82.30 |
| *AIDE* | 79.38 | 99.74 | 99.76 | **78.54** | **91.82** | 98.65 | 80.26 | 66.89 | **86.88** |

extract semantic features. We adopt data augmentations including random JPEG compression (QF $\sim$ Uniform$(30, 100)$) and random Gaussian blur ($\sigma \sim$ Uniform$(0.1, 3.0)$) to improve the robustness of detectors. Each augmentation is conducted with 10% probability. During the training phase, we use AdamW optimizer with the learning rate of $1 \times 10^{-4}$ in $\mathcal{B}_1$ and $5 \times 10^{-4}$ in $\mathcal{B}_2$, respectively. The batch size is set to 32 and the model is trained on 8 NVIDIA A100 GPUs for only 5 epochs. Our method trains very quickly, only 2 hours are sufficient.

**Metrics.** In accordance with existing AI-generated detection approaches Wang et al. (2020; 2019); Zhou et al. (2018), we report both classification accuracy (Acc) and average precision (AP) in our experiments. All results are averaged over both real and AI-generated images unless otherwise specified. We primarily report Acc for evaluation and comparison in the main paper, and AP results are presented in the Appendix.

## 5.2 COMPARISON TO STATE-OF-THE-ART MODELS

**On Benchmark AIGCDetectBenchmark**: The quantitative results in Table 3 present the classification accuracies of various methods and generators within $\mathcal{B}_1$. In this evaluation, all methods, except for DIRE-D, were exclusively trained on ProGAN-generated data.

AIDE demonstrates a significant advancement over the existing state-of-the-art (SOTA) approach, for example, PatchCraft, achieving an average accuracy increase of 3.5%. UnivFD utilizes CLIP semantic features for detecting AI-generated images, proving effective for GAN-generated images. However, it shows pronounced performance degradation with diffusion model (DM)-generated images. This suggests that as generation quality improves, diffusion models produce images with fewer semantic artifacts, as depicted in Fig. 1 (a). Our approach, which integrates semantic, low-frequency, and high-frequency information at the feature level, enhances detection performance, yielding a 5.2% increase for GAN-based images and a 1.7% increase for DM-based images compared to the SOTA method.

Table 5: **Comparison on the `Chameleon`.** Accuracy (%) of different detectors (rows) in detecting real and fake images of `Chameleon`. For each training dataset, the first row indicates the **Acc** evaluated on the `Chameleon` testset, and the second row gives the **Acc** for "**fake image / real image**" for detailed analysis.

| Training Dataset | CNNSpot | FreDect | Fusing | GramNet | LNP | UnivFD | DIRE | PatchCraft | NPR | AIDE |
|---|---|---|---|---|---|---|---|---|---|---|
| **ProGAN** | 56.94 | 55.62 | 56.98 | **58.94** | 57.11 | 57.22 | 58.19 | 53.76 | 57.29 | 58.37 |
| | 0.08/99.67 | 13.72/87.12 | 0.01/99.79 | 4.76/99.66 | 0.09/99.97 | 3.18/97.83 | 3.25/99.48 | 1.78/92.82 | 2.20/98.70 | 5.04/98.46 |
| **SD v1.4** | 60.11 | 56.86 | 57.07 | 60.95 | 55.63 | 55.62 | 59.71 | 56.32 | 58.13 | **62.60** |
| | 8.86/98.63 | 1.37/98.57 | 0.00/99.96 | 17.65/93.50 | 0.57/97.01 | 74.97/41.09 | 11.86/95.67 | 3.07/96.35 | 2.43/100.00 | 20.33/94.38 |
| **All GenImage** | 60.89 | 57.22 | 57.09 | 59.81 | 58.52 | 60.42 | 57.83 | 55.70 | 57.81 | **65.77** |
| | 9.86/99.25 | 0.89/99.55 | 0.02/99.98 | 8.23/98.58 | 7.72/96.70 | 85.52/41.56 | 2.09/99.73 | 1.39/96.52 | 1.68/100.00 | 26.80/95.06 |

Table 6: **Robustness on JPEG compression and Gaussian blur of AIDE.** The accuracy (%) averaged over 16 test sets in $\mathcal{B}_1$ with specific perturbation.

Table 7: **Ablation studies on PFE and SFE modules.**

| Method | Original | JPEG Compression | | | | Gaussian Blur | | | |
|---|---|---|---|---|---|---|---|---|---|
| | | QF=95 | QF=90 | QF=75 | QF=50 | $\sigma = 1.0$ | $\sigma = 2.0$ | $\sigma = 3.0$ | $\sigma = 4.0$ |
| CNNSpot | 70.78 | 64.03 | 62.26 | 60.65 | 59.66 | 68.39 | 67.26 | 67.13 | 65.85 |
| FreDect | 64.03 | 66.95 | 67.45 | 66.64 | 65.33 | 65.75 | 66.48 | 68.58 | 69.64 |
| Fusing | 68.38 | 62.43 | 61.39 | 59.34 | 57.41 | 68.09 | 66.69 | 66.02 | 65.58 |
| LNP | 83.84 | 53.58 | 54.09 | 53.02 | 52.85 | 67.91 | 66.42 | 66.2 | 62.69 |
| LGrad | 75.34 | 51.55 | 51.39 | 50.00 | 50.00 | 71.73 | 69.12 | 68.43 | 66.22 |
| DIRE-G | 68.68 | 66.49 | 66.12 | 65.28 | 64.34 | 64.00 | 63.09 | 62.21 | 61.91 |
| UnivFD | 78.43 | 74.10 | 74.02 | 69.92 | 68.68 | 70.31 | 68.29 | 64.62 | 61.18 |
| PatchCraft | 89.31 | 72.48 | 71.41 | 69.43 | 67.78 | 75.99 | 74.90 | 73.53 | 72.28 |
| AIDE | **92.77** | **75.54** | **74.21** | **70.64** | **69.60** | **81.88** | **80.35** | **80.05** | **79.86** |

| | Module | | | Mean |
|---|---|---|---|---|
| PFE-H | PFE-L | SFE | | |
| ✓ | ✗ | ✗ | | 76.09 |
| ✗ | ✓ | ✗ | | 75.24 |
| ✗ | ✗ | ✓ | | 75.26 |
| ✓ | ✓ | ✗ | | 76.70 |
| ✓ | ✗ | ✓ | | 80.69 |
| ✗ | ✓ | ✓ | | 84.20 |
| ✓ | ✓ | ✓ | | **92.77** |

**On Benchmark GenImage**: In the experiments conducted on $\mathcal{B}_2$, all models were trained on SD v1.4 and evaluated across eight contemporary generators. Table 4 presents the results, illustrating our method's superior performance over the current state-of-the-art, PatchCraft, with a 4.6% improvement in average accuracy. The architectural similarities between SD v1.5, Wukong, and SD v1.4, as noted by GenImage (Zhu et al., 2024), enable models to achieve near-perfect accuracy, approaching 100% on such datasets. Thus, evaluating generalization performance across other generators, such as Midjourney, ADM, and Glide, becomes essential. Our model demonstrates either the best or second-best performance in these cases, achieving an average accuracy of 86.88%.

**On Benchmark `Chameleon`**: As highlighted in Sec. 1, we contend that success on existing public benchmarks may not accurately reflect real-world scenarios or the advancement in AI-generated image detection, given that test sets are typically randomly sampled from generative models without "Turing Test". To address potential biases related to training setups—such as generator types and image categories—we evaluate the performance of existing detectors under diverse training conditions. Despite their high performance on existing benchmarks, as depicted in Fig. 8, the state-of-the-art detector, PatchCraft, experiences substantial performance declines. Additionally, Table 5 reveals significant performance decreases across all methods, with most struggling to surpass an average accuracy close to random guessing (about 50%), indicating a failure in these contexts.

While our method achieves the SOTA results on available datasets, its performance on `Chameleon` remains low, especially on discovering the AI-generated images. This underscores that our dataset, `Chameleon`, which challenges human perception, is also extremely difficult AI models.

## 5.3 ROBUSTNESS TO UNSEEN PERTURBATIONS

In real-world scenarios, images inevitably encounter unseen perturbations in transmission and interaction, complicating AI-generated image detection. Herein, we assess the performance of various methods in handling potential perturbations, such as JPEG compression (Quality Factor (QF) = 95, 90, 75, 50) and Gaussian blur ($\sigma$ = 1.0, 2.0, 3.0, 4.0). As illustrated in Table 6, all methods exhibit a decline in performance due to disruptions in the pixel distribution. These disruptions diminish the discriminative artifacts left by generative models, complicating the differentiation between real and AI-generated images. Consequently, the robustness of these detectors in identifying AI-generated images is significantly compromised. Despite these challenging conditions, our method consistently outperforms others, maintaining a relatively higher accuracy in detecting AI-generated images. This superior performance is attributed to our model's ability to effectively capture and leverage multi-perspective features, semantics, and noise, even when the pixel distribution is distorted.

 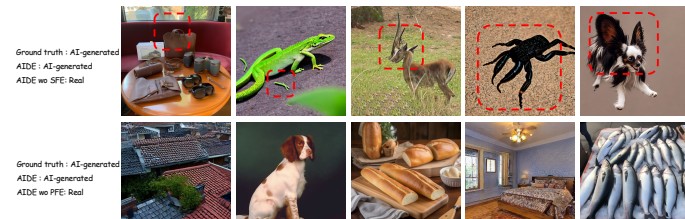

Table 8: **Performance of SOTA method, PatchCraft** Zhong et al. (2023), **under** $\mathcal{B}_1$ **(left),** $\mathcal{B}_2$ **(right), and our** `Chameleon` **testset.** The boundary line for Acc = 50% is marked with a dashed line.

Table 9: **Visualization of the effectiveness of PFE and SFE modules.** Without SFE, AI-generated images exhibit pronounced semantic errors that are incorrectly classified as real. Similarly, without PFE, many AI-generated images, despite lacking semantic errors, contain subtle underlying noise that also leads to their misclassification as true.

## 5.4 ABLATION STUDIES

Our method focuses on detecting AI-generated images with mixture of experts, namely patchwise feature extraction (PFE-H and PFE-L for high-frequency and low-frequency patches, respectively) and semantic feature extraction (SFE). These modules collectively contribute to comprehensively identifying AI-generated images from different perspectives. Herein, we conduct extensive experiments to investigate the roles of each module on $\mathcal{B}_1$.

**Patchwise Feature Extraction.** As shown in Table 7, removing either the high-frequency or the low-frequency patches results in obvious performance degradation in terms of accuracy. Without the high-frequency patches, the proposed method is unable to discern that the high-frequency regions of AI-generated images are smoother than those of real images, resulting in performance degradation. Similarly, without the low-frequency patches, the method cannot extract the underlying noise information, which is crucial for identifying AI-generated images with higher fidelity, leading to incorrect predictions.

**Semantic Feature Extraction.** The performance degrades significantly (76.70% vs 92.77%) when we remove the semantic branch, as shown in Table 7. Intuitively, if the branch for semantic information extraction is absent, our method struggles to effectively capture images with semantic artifacts, resulting in significant drops.

**Visualization.** To vividly demonstrate the effectiveness of our modules patchwise feature extraction (PFE) and semantic feature extraction (SFE), we conducted a visualization, as depicted in Fig. 9. In the first row, the absence of semantic feature extraction results in many images with evident semantic errors going undetected. Similarly, the second row shows that, without patchwise feature extraction, numerous images lacking semantic errors still contain differing underlying information that remains unrecognized. Overall, our method, AIDE, achieves the best performance.

## 6 CONCLUSION

In this paper, we have conducted a sanity check on detecting AI-generated images. Specifically, we re-examined the unreasonable assumption in existing training and testing settings and suggested new ones. In terms of benchmarks, we propose a novel, challenging benchmark, termed as `Chameleon`, which is manually annotated to challenge human perception. We evaluate 9 off-the-shelf models and demonstrate that all detectors suffered from significant performance declines. In terms of architecture, we propose a simple yet effective detector, **AIDE**, that simultaneously incorporates low-level patch statistics and high-level semantics for AI-generated image detection. Despite our approach demonstrates state-of-the-art performance on existing (AIGCDetectBenchmark (Zhong et al., 2023) and GenImage (Zhu et al., 2024)) and our proposed benchmark (`Chameleon`) compared to previous detectors, it leaves significant room for future improvement.

## ACKNOWLEDGEMENTS

WX would like to acknowledge the National Key R&D Program of China (No. 2022ZD0161400).

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

APPENDIX

## A EXPERIMENTAL DETAILS

### A.1 DETECTORS

We choose a set of representative methods in AI-generated detection as baselines for comparison, including frequency-based (Frank et al., 2020; Liu et al., 2022a; Zhong et al., 2023), gradient-based (Tan et al., 2023), semantic-based (Ojha et al., 2023), reconstruction-based (Wang et al., 2023a), etc.

- CNNSpot (CVPR'2020) (Wang et al., 2020) proposes that a naïve image classifier with simple data augmentations (*i.e.,* JPEG compression and Gaussian blur) can generalize surprisingly to images generated by unknown GAN-based architectures.
- FreDect (ICML'2020) (Frank et al., 2020) observes significant artifacts in the frequency domain of GAN-generated images and makes use of these artifacts for classification.
- Fusing (ICIP'2022) (Ju et al., 2022) designs a two-branch model to fuse global spatial information and local informative features for training the classifier.
- LNP (ECCV'2022) (Liu et al., 2022a) proposes to extract the noise pattern of images with a learnable denoising network and uses noise patterns to train a classifier.
- LGrad (CVPR'2023) (Tan et al., 2023) employs gradients computed by a pretrained CNN model to present the generalized artifacts for classification.
- UnivFD (CVPR'2023) (Ojha et al., 2023) uses CLIP features to train a binary liner classifier.
- DIRE (ICCV'2023) (Wang et al., 2023a) observes obvious differences in discrepancies between images and their reconstruction by DMs and uses this feature to train a classifier.
- PatchCraft (Arxiv'2024) (Zhong et al., 2023) compares rich-texture and poor-texture patches from images and extracts the inter-pixel correlation discrepancy as a universal fingerprint for classification.
- NPR (CVPR'2024) (Tan et al., 2024) contributes to the architectures of CNN-based generators and demonstrates that the up-sampling operator can generate generalized forgery artifacts that extend beyond mere frequency-based artifacts.

### A.2 STATISTICS OF PUBLIC BENCHMARKS

Table 10 and Table 11 provide a detailed explanation of Benchmark AIGCDetectBenchmark (Zhong et al., 2023) & GenImage (Zhu et al., 2024) introduced in our main paper. There are two main benchmarks here: AIGCDetectBenchmark and GenImage. **AIGCDetectBenchmark**: It is trained on ProGAN and then tested on 16 different test sets, including data generated by both GAN and Stable Diffusion models. **GenImage**: It is trained on Stable Diffusion V1.4 and tested on a large amount of data generated by Stable Diffusion, with only a small portion of GAN data included. The test sets related to Stable Diffusion in AIGCDetectBenchmark are consistent with those used in GenImage.

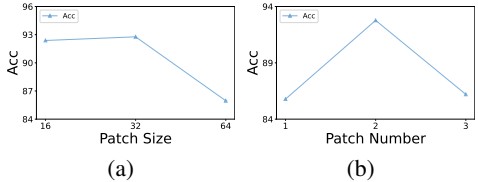

Figure 3: Hyperparameter ablation of patch size and patch number introduced in our method.

## B MORE EXPERIMENTAL RESULTS

### B.1 AP RESULT

We additionally provide classification results regarding AP in Table 12. It is important to highlight that the AP (Average Precision) metric emphasizes different aspects compared to Acc (Accuracy). While Acc focuses on the overall correctness of predictions across all samples, AP provides a more

Table 10: **Statistics of Benchmark AIGCDetctBenchmark.** SD and WFIR refer to Stable Diffusion and whichfaceisreal, respectively. The term "Number" only counts on fake images and an equal number of real images is added for each generative model from the same source.

| | Generator | Image Size | Number | Source |
|---|---|---|---|---|
| **Train** | ProGAN (Karras et al., 2018) | $256 \times 256$ | 360.0k | LSUN (Yu et al., 2015) |
| **Test** | ProGAN (Karras et al., 2018) | $256 \times 256$ | 8.0k | LSUN (Yu et al., 2015) |
| | StyleGAN (Karras et al., 2019) | $256 \times 256$ | 12.0k | LSUN (Yu et al., 2015) |
| | BigGAN (Brock et al., 2019) | $256 \times 256$ | 4.0k | ImageNet (Deng et al., 2009) |
| | CycleGAN (Zhu et al., 2017) | $256 \times 256$ | 2.6k | ImageNet (Deng et al., 2009) |
| | StarGAN (Choi et al., 2018) | $256 \times 256$ | 4.0k | CelebA (Liu et al., 2015) |
| | GauGAN (Park et al., 2019) | $256 \times 256$ | 10.0k | COCO (Lin et al., 2014) |
| | StyleGAN2 (Karras et al., 2020) | $256 \times 256$ | 15.9k | LSUN (Yu et al., 2015) |
| | WFIR (WFI, 2023) | $1024 \times 1024$ | 2.0k | FFHQ (Karras et al., 2019) |
| | ADM (Dhariwal & Nichol, 2021) | $256 \times 256$ | 12.0k | ImageNet (Deng et al., 2009) |
| | Glide (Nichol et al., 2022) | $256 \times 256$ | 12.0k | ImageNet (Deng et al., 2009) |
| | Midjourney (mid) | $1024 \times 1024$ | 12.0k | ImageNet (Deng et al., 2009) |
| | SD v1.4 (Sta, 2022) | $512 \times 512$ | 12.0k | ImageNet (Deng et al., 2009) |
| | SD v1.5 (Sta, 2022) | $512 \times 512$ | 16.0k | ImageNet (Deng et al., 2009) |
| | VQDM (Gu et al., 2022) | $256 \times 256$ | 12.0k | ImageNet (Deng et al., 2009) |
| | Wukong (wuk, 2023) | $512 \times 512$ | 12.0k | ImageNet (Deng et al., 2009) |
| | DALLE 2 (Ramesh et al., 2022) | $256 \times 256$ | 2.0k | ImageNet (Deng et al., 2009) |

Table 11: **Statistics of Benchmark GenImage.** SD refers to Stable Diffusion. The term "Number" only counts on fake images and an equal number of real images is added for each generative model from the same source.

| | Generator | Image Size | Number | Source |
|---|---|---|---|---|
| **Train** | SD v1.4 (Sta, 2022) | $512 \times 512$ | 324.0k | ImageNet Deng et al. (2009) |
| **Test** | BigGAN (Brock et al., 2019) | $256 \times 256$ | 12.0k | ImageNet (Deng et al., 2009) |
| | ADM (Dhariwal & Nichol, 2021) | $256 \times 256$ | 12.0k | ImageNet (Deng et al., 2009) |
| | Glide (Nichol et al., 2022) | $256 \times 256$ | 12.0k | ImageNet (Deng et al., 2009) |
| | Midjourney (mid) | $1024 \times 1024$ | 12.0k | ImageNet (Deng et al., 2009) |
| | SD v1.4 (Sta, 2022) | $512 \times 512$ | 12.0k | ImageNet (Deng et al., 2009) |
| | SD v1.5 (Sta, 2022) | $512 \times 512$ | 16.0k | ImageNet (Deng et al., 2009) |
| | VQDM (Gu et al., 2022) | $256 \times 256$ | 12.0k | ImageNet (Deng et al., 2009) |
| | Wukong (wuk, 2023) | $512 \times 512$ | 12.0k | ImageNet (Deng et al., 2009) |

comprehensive evaluation of a model's performance across various thresholds, particularly in handling imbalanced datasets. On top of that, our method still achieves SOTA performance among these baselines on AP metric, which underscores the superiority of our approach. This indicates that our method not only excels in general prediction accuracy but also maintains robust performance across different decision thresholds, demonstrating its effectiveness in distinguishing between classes even in challenging scenarios.

## B.2 MORE ABLATION STUDIES

### B.2.1 PATCH NUMBER AND PATCH SIZE

We further ablate some key parameters defined in our method, namely the size of patches (Patch Size) and the num of selected patches (Patch Number). As shown in Fig. 3 (a), both an excessively large and an overly small patch size can have certain impacts. If the patch size is too large, it may introduce additional irrelevant information, leading to interference with accurate judgment. On the other hand, if the patch size is too small, there may not be enough information available to make a proper judgment. As shown in Fig. 3 (b), for the patch number, the conclusion is that there is a correlation between the number of patches and the patch size.

Table 12: **Comparison on the AIGCDetectBenchmark (Zhong et al., 2023) benchmark.** Average precision (AP %) of different detectors (rows) in detecting real and fake images from different generators (columns). The best result and the second-best result are marked in **bold** and underline, respectively.

| Method | ProGAN | StyleGAN | BigGAN | CycleGAN | StarGAN | GauGAN | StyleGAN2 | WFIR | ADM | Glide | Midjourney | SD v1.4 | SD v1.5 | VQDM | Wukong | DALLE2 | Mean |
|---|---|---|---|---|---|---|---|---|---|---|---|---|---|---|---|---|---|
| CNNSpot | **100.00** | 99.83 | 85.99 | 94.94 | 99.04 | 90.82 | 99.48 | **99.85** | 75.67 | 72.28 | 66.24 | 61.20 | 61.56 | 68.83 | 57.34 | 53.51 | 80.41 |
| FreDect | 99.99 | 88.98 | 93.62 | 84.78 | 99.49 | 82.84 | 82.54 | 55.85 | 61.77 | 52.92 | 46.09 | 37.83 | 37.76 | 85.10 | 39.58 | 38.20 | 67.96 |
| Fusing | **100.00** | 99.50 | 90.70 | 95.50 | 99.80 | 88.30 | 99.60 | 93.30 | 94.10 | 77.50 | 70.00 | 65.40 | 65.70 | 75.60 | 64.60 | 68.12 | 84.23 |
| LNP | 99.89 | 98.60 | 84.32 | 92.83 | **100.00** | 78.85 | 99.59 | 91.45 | 94.20 | 88.86 | 76.86 | 94.31 | 93.92 | 87.35 | 92.38 | 96.14 | 91.85 |
| LGrad | **100.00** | 98.31 | 92.93 | 95.01 | **100.00** | 95.43 | 97.89 | 57.99 | 72.95 | 80.42 | 71.86 | 62.37 | 62.85 | 77.47 | 62.48 | 82.55 | 81.91 |
| UnivFD | 99.08 | 91.74 | 75.25 | 80.56 | 99.34 | 72.15 | 88.29 | 60.13 | 85.84 | 78.35 | 61.86 | 49.87 | 49.52 | 54.57 | 55.38 | 74.48 | 73.53 |
| DIRE-G | 58.79 | 56.68 | 46.91 | 50.03 | 40.64 | 47.34 | 58.03 | 59.02 | **99.79** | **99.54** | **97.32** | 98.61 | 98.83 | 98.98 | 98.37 | **99.71** | 75.54 |
| DIRE-D | **100.00** | 97.56 | 99.27 | 99.80 | 99.37 | **99.98** | 97.90 | 96.73 | 86.81 | 83.81 | 74.00 | 86.14 | 85.84 | 96.53 | 91.07 | 63.04 | 91.12 |
| PatchCraft | **100.00** | 98.96 | **99.42** | 85.26 | **100.00** | 81.33 | 97.74 | 95.26 | 93.40 | 94.04 | 96.48 | **99.06** | **99.06** | 96.26 | 97.54 | 99.56 | 95.84 |
| NPR | **100.00** | 99.81 | 87.87 | 98.55 | 99.90 | 85.57 | 99.90 | 65.38 | 74.61 | 85.73 | 85.41 | 84.02 | 84.67 | 81.20 | 80.51 | 76.72 | 86.87 |
| *AIDE* | **100.00** | **99.99** | 94.44 | **99.89** | 99.99 | 97.69 | **99.96** | 99.27 | 98.77 | 98.94 | 88.13 | 98.26 | 98.20 | **99.27** | **98.62** | 99.41 | **98.18** |

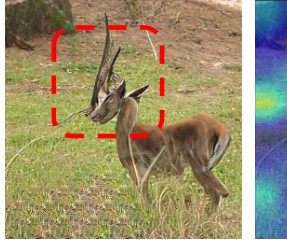 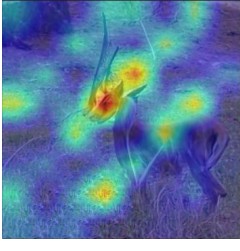 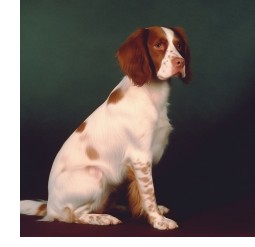 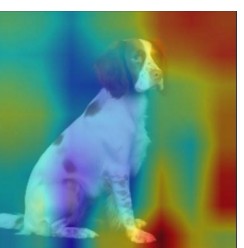

(a). Ground truth : AI-generated
AIDE : AI-generated
AIDE wo SFE: Real

(b). Ground truth : AI-generated
AIDE : AI-generated
AIDE wo PFE: Real

Figure 4: **Visualization of the effectiveness of PFE and SFE Modules with Grad-CAM (Selvaraju et al., 2017).**

### B.2.2 CONVNEXT AND VIT

We further explored the impact of CNN-Supervised network architectures such as CLIP-ConvNeXt and ViT-Supervised CLIP-ViT. The results of CLIP-ViT on **Benchmark 1** have an average accuracy of 80.87%, which is significantly lower than the 92.77% achieved by CLIP-ConvNeXt. We speculate that this could be due to the CNN-supervised network architecture learning more low-level information, and for AI-generated image detection, low-level information is most crucial when the images are highly realistic.

## C LIMITATIONS

Although our method achieves state-of-the-art results on publicly available datasets and demonstrates highly competitive performance on **Chameleon**, its performance on our own dataset is still unsatisfactory. It leaves significant room for future improvement. Additionally, while the quality of our dataset is exceptionally high, its scale remains limited, and we aim to further expand its scale to better facilitate advancements in this field.

## D VISUALIZATION

To more effectively verify the efficacy of the patchwise feature extraction (PFE) and semantic feature extraction (SFE) modules, we employ Grad-CAM (Selvaraju et al., 2017) to visualize the feature areas targeted by these modules. In Figure 4 (a), it is evident that the region highlighted by the red box exhibits distinct semantic issues, which our SFE module successfully captures with clarity. Conversely, in Figure 4 (b), there are no apparent semantic errors, and our PFE module accurately detects the low-level underlying noise information. Overall, our model, AIDE, demonstrates outstanding performance as a detector.

# E  POTENTIAL SOCIETAL IMPACTS

Given that **Chameleon** demonstrates the capability to surpass the "Turing Test", there exists a significant risk of exploitation by malicious entities who may utilize AI-generated imagery to engineer fictitious social media profiles and propagate misinformation. To mitigate it, we will require all users of **Chameleon** to enter into an End-User License Agreement (EULA). Access to the dataset will be contingent upon a thorough review and subsequent approval of the signed agreement, thereby ensuring compliance with established ethical usage protocols.

