# OpenReview forum: "A Sanity Check for AI-generated Image Detection"
_ICLR.cc/2025/Conference — ICLR 2025 Poster_

### Official Review · Reviewer_9Ryx · 2024-10-28

**Soundness:** 3
**Presentation:** 3
**Contribution:** 2
**Rating:** 6
**Confidence:** 5

**Summary:**

This paper introduces the Chameleon dataset and the AIDE detector for distinguishing between AI-generated and real images.

**Strengths:**

S1: Comprehensive Dataset: The Chameleon dataset offers a large-scale and diverse collection of AI-generated and real images, providing a robust foundation for training and evaluating detection models.

S2: Detailed Performance Evaluation: The paper provides thorough baseline comparisons and benchmarking, highlighting the strengths and weaknesses of the AIDE detector across various scenarios.

S3: Practical Applications: The focus on detecting AI-generated images is highly relevant in today's digital landscape, with the paper identifying practical areas for improvement.

**Weaknesses:**

W1: Method Innovation: The combination of patch-level features and global features proposed in this paper is not novel. The methods compared are mostly from the first half of 2023 and earlier, and some of the latest work is not included (such as LaRE^2, Forgery-aware Adaptive Transformer, etc.).

W2: Inference Speed: The inference throughput of the proposed method is not provided, nor is it compared with previous methods. The AIDE uses multiple SRMs, ResNet50, and CLIP/ConvNext models, which are much larger than existing work like Gendet that uses just ResNet-50 as the classifier. This raises questions about whether the performance increase claimed in the article is due to increased FLOPS/PARAMS or the AIDE itself.

W3: Dataset Comparison: The Chameleon dataset has not been comprehensively compared with existing datasets (such as CNNSpot, AIGCDetect, GenImage, WildFake, etc.) in terms of data size, data sources, semantic categories, generation models, image quality, and other relevant factors. Its novelty has not been sufficiently demonstrated.

**Questions:**

Q1: What is the cost of data collection, and is it feasible to annotate more data to scale up the dataset?

---

> ### Author Response · Authors · 2024-11-21
>
> We appreciated the insightful reviews and valuable suggestion. We hope the following response can address your concerns.
>
> > **Q1. The combination of patch-level features and global features proposed is not novel.**
>
> Although low-level information and global features are common, we have carefully designed components to address significant challenges in AI-generated image detection, which have been verified by many benchmarks, e.g., AIGCDetectBenchmark, GenImage, and Chameleon. Our research contributions are fourfold:
>
> **1. Patchwise Feature Extraction**. We designed a **block scoring mechanism** to selectively highlight the highest and lowest frequency blocks. Our patchwise feature extraction module targets disparities in underlying information, such as the presence of white noise during image capturing.
>
> **2. Semantic  Feature Embedding**. Our semantic feature embedding module captures object co-occurrence and contextual relationships. For instance, penguins are unlikely to appear on the grasslands of Africa.
>
> **3. Generalization on different settings**.  The significant improvement over previous SOTA methods in various settings demonstrates our method's strong generalization capability.
>
> **4. Visualization**. As visualized in ***Fig 4 of main paper***, without semantic feature extraction, which results in many images with evident semantic errors going undetected. Similarly, without patchwise feature extraction, numerous images lacking semantic errors still contain differing underlying information that remains unrecognized.
>
> Therefore, we believe that our method is novel and valuable. In addition, we present a more challenging dataset characterized by comprehensive manual annotation, real-world applicability, high resolution, and large scale.
>
> > **Q2. The comparsion of recent works.**
>
> Thanks for the suggestion.  We have included new baseline model, namely, FatFormer, and use their publicly available model for evaluation. The detailed comparison is as follows and we will add them in revised version.
>
> Notably: **LaRE².** However, we discovered that the LaRE² method was open-sourced only in the last three weeks and lacks training weights. In our attempts to replicate the results, we encountered missing essential code files, an issue that has also been noted by others in the GitHub issues section.
>
>
> The table below shows the results of ***FatFormer*** and ***AIDE*** on AIGCDetectBenchmark. As shown in the table below, ***AIDE*** surpasses ***FatFormer*** by a large margin.
>
> ||ProGAN|StyleGAN|BigGAN|CycleGAN|StarGAN|GauGAN|Stylegan2|WFIR|ADM|Glide|Midjourney|SDv1.4|SDv1.5|VQDM|Wukong|DALLE2|Avg|
> |:-:|:-:|:-:|:-:|:-:|:-:|:-:|:-:|:-:|:-:|:-:|:-:|:-:|:-:|:-:|:-:|:-:|:-:|
> |FatFormer|99.89|97.15|99.50|99.32|100.00|99.41|98.80|88.05|78.40|87.98|56.07|67.79|68.03|86.86|73.02|69.70|85.62|
> |AIDE|99.99|99.64|83.95|98.48|99.91|73.25|98.00|94.20|93.45|95.09|77.20|93.00|92.85|95.16|93.55|96.60|**92.77**|
>
> > **Q3. The inference throughput of the proposed method is not provided, nor is it compared with previous methods.**
>
> Thanks for pointing this out. We further evaluate throughput of the proposed method and recent works.
>
> |Method|Ref|FLOPs|AIGCDetectBenchmark|
> |:-:|:-:|:-:|:-:|
> |UnivFD|CVPR 2023|77.83B|78.43|
> |FatFormer|CVPR 2024|127.95B|85.91|
> |AIDE|-|225.88B|92.77|
>
> > **Q4. Whether the performance increase claimed in the article is due to increased FLOPS/PARAMS or the AIDE itself.**
>
> We further conducted additional experiments to explore the core components of AIDE.
>
> **1. Exploring the *Patchwise Feature Extraction* in our AIDE**:  We replaced the high and low-frequency blocks with the original images and conducted experiments on AIGCDetectBenchmark. As shown below, our high and low-frequency blocks are extremely effective.
>
> |Method|AIGCDetectBenchmark|
> |:-:|:-:|
> |Replace High and Low|75.95|
> |AIDE|92.77|
>
> **2. Ablation Study.** The CLIP-ConvNeXt occupies 94% of the parameters in AIDE. When using only CLIP-ConvNeXt, there is a significant drop in performance on AIGCDetectBenchmark, as shown below. This demonstrates that our method's effectiveness is not due to an increase in the number of parameters.
>
> |Method|AIGCDetectBenchmark|Parameters %|
> |:-:|:-:|:-:|
> |Only CLIP-ConvNeXt|75.26|94.05|
> |AIDE|92.77|100|

---

> > ### Author Response · Authors · 2024-11-21
> >
> > > **Q5. The Chameleon dataset has not been comprehensively compared with existing datasets and the novelty has not been sufficiently demonstrated.**
> >
> > Sorry for the confusion. We will clarify this with more discussions in the revised version.
> >
> > 1. **Quantitative comparison**. We conducted a comprehensive comparison of the Chameleon dataset with existing test sets, as depicted in ***Tab 2 of the main paper***. Additionally, a detailed analysis is provided in ***L254–262 of main paper***. Chameleon is a challenging dataset characterized by comprehensive manual annotation, real-world applicability, high resolution, and large scale.
> >
> > 2. **Qualitative comparison**. ***Fig 1 of the main paper*** visually compares our dataset with other existing datasets. It is evident that existing benchmarks exhibit noticeable flaws (highlighted ***in the red box in Fig 1***), whereas our dataset maintains high fidelity.
> >
> >
> > The unique contribution of our Chameleon lies in the following five aspects:
> > 1. **Deceptively real.**: All AI-generated images in the dataset have passed a human perception ***"Turing Test"***, i.e., human annotators have misclassified them as real images.
> > 2. **Diversity.**: The Chameleon dataset is derived from ***real-world scenarios***, ensuring a broad and authentic representation. Notably, the dataset exhibits wide distribution, for example, with data generated by a diverse range of users utilizing LoRA (Low-Rank Adaptation) technology.
> > 3. **High resolution.**: With most images having resolutions exceeding ***720P*** and going up to ***4K***, all images in the dataset exhibit exceptional clarity. Consequently, this test set offers a more realistic evaluation of model performance. However, existing datasets show low resolution.
> > 4. **Large-Scale.** As illustrated in ***Tab 2 of the main paper***, our current Chameleon dataset stands out as the largest available.
> > 5. **Valueable.** The dataset holds significant value. As indicated in ***Tab 5 of the main paper***, all existing detection methods are rendered ineffective. Conversely, as presented in ***Tab 3 and 4***, existing test datasets demonstrate that most detectors perform well, with some achieving near-perfect accuracy.
> >
> > Consequently, our Chameleon dataset serves as a new real-world benchmark that can advance the development of AI-generated image detection.
> >
> >  > **Q6. What is the cost of data collection, and is it feasible to annotate more data to scale up the dataset?**
> >
> > 1. **Cost.** We invested approximately ***3,000 human hours*** in the labeling process. This process is labor-intensive, as each image is independently labeled by two annotators. Labels are retained only when both annotators agree.
> >
> > 2. **Scale up the dataset**. We believe the current size of the Chameleon dataset is adequate for use as a test set, as demonstrated by the comparison with other benchmarks in ***Tab 2 of the main paper***. Additionally, we plan to annotate more data in future revisions.

---

> > > ### Comment · Reviewer_9Ryx · 2024-11-26
> > >
> > > I would like to thank the author for his answers to my questions. However, I do not agree with the novelty of the proposed method. The application of the patch had proposed in [https://github.com/bcmi/SSP-AI-Generated-Image-Detection](https://github.com/bcmi/SSP-AI-Generated-Image-Detection), this paper simply joining together. Given that the dataset is valuable for the community to explore high-resolution detection, I can improve my score.

---

> > > > ### Author Response · Authors · 2024-11-26
> > > > **Thanks for your recognition of  our work**
> > > >
> > > > Dear Reviewer 9Ryx,
> > > >
> > > > We express our gratitude for your acknowledgment of the significance of our dataset. We will provide further clarification on the method novelty in our revised version.
> > > >
> > > > Once again, we appreciate your recognition of the importance of our dataset.

---

> > > > > ### Public Comment · ~Li_Niu2 · 2024-11-29
> > > > >
> > > > > This is a great work. It is suggested to cite and discuss SSP https://github.com/bcmi/SSP-AI-Generated-Image-Detection.

---

### Official Review · Reviewer_QHiA · 2024-11-03

**Soundness:** 2
**Presentation:** 3
**Contribution:** 2
**Rating:** 6
**Confidence:** 4

**Summary:**

In this paper, the authors propose a new AI generated image dataset collected from the real-life scenarios, called Chameleon, in which the forgery images are genuinely challenging for human perception. Besides, this paper also provides a novel detection method, which exhibits better detection performance than previous SOTA methods.

**Strengths:**

•	This paper provides a challenging AI generated image dataset, which is collected from real-life scenarios, and indicates that the problem for detecting AI-generated images is far from being solved.
•	The authors provide a new detection method, achieving better performance than previous methods.
•	This paper is organized well.

**Weaknesses:**

•	The rationale behind how patches with the highest and lowest frequency highlight the distinctive characteristics of AI-generated images is unclear. The manuscript lacks a theoretical explanation for the proposed method.

•	In Table 2, comparing the entire dataset with subsets of the AIGCDetectBenchmark dataset seems unreasonable. A more appropriate comparison framework should be established.

•	Figure 4 does not provide sufficient evidence to support the claim that SFE (or PFE) can assist AIDE in focusing on specific semantic (or noise-related) clues. It may be beneficial to employ Grad-CAM to visualize the regions of interest when detecting forgery in images.

**Questions:**

•	The questions are as mentioned above.

---

> ### Author Response · Authors · 2024-11-21
>
> We sincerely appreciate your careful reviews and constructive comments. We hope our response can address your concerns.
>
> > **Q1. Theoretical explanation for the proposed method.**
>
>
> 1. **High Frequency Patches.** Generative models often produce ***smoother patches*** in high-frequency areas, such as when generating hair. As a result, image patches in these regions can serve as a powerful tool for identifying AI-generated images.
>
> 2. **Low Frequency Patches.**  In low-frequency areas, like patches of solid color, a generative model primarily needs to match the surrounding color. However, when a camera captures real images, achieving perfectly uniform color is very unlikely due to the presence of ***various types of noise***.
>
> > **Q2. Comparing the entire dataset with subsets of the AIGCDetectBenchmark dataset seems unreasonable.**
>
> Sorry for the confusion. We do not mean to compare our dataset with existing ones. Our goal is to show that our dataset serves as a valuable extension of existing AI-generated image detection evaluation.
> 1. **Deceptively real.** As far as we know, this is the first dataset containing images that are manually curated, which are indistinguishable from human observers.
> 2. **Diversity.** The Chameleon dataset is derived from ***real-world scenarios***, ensuring a broad and authentic representation.
> 3. **High resolution.** Considering that the majority of images have resolutions surpassing ***720P*** and reaching up to ***4K**, the entire dataset exhibits exceptional clarity. And this high resolution poses increased challenges for the identification of artifacts.
> 4. **Large-Scale.** As illustrated in ***Tab 2 of the main paper***, our current Chameleon dataset stands out as the largest available.
>
> Consequently, both high- and low-frequency patches are valuable in the detection of AI-generated images.
>
> > **Q3. It may be beneficial to employ Grad-CAM to visualize the regions of interest when detecting forgery in images.**
>
> Thanks for the advice. We have incorporated new visualization in ***Figure 6***, located in the ***appendix of the revised manuscript***. Additionally, we have included a more comprehensive analysis in ***L941–949 of the main paper***.

---

> > ### Comment · Reviewer_QHiA · 2024-11-25
> >
> > Thanks for the responses.
> >
> > About comparing to AIGCDetectBenchmark subsets, more specifically, I mean "Chameleon offers greater magnitude" is not correct. The proposed dataset is smaller than AIGCDetectBenchmark as a whole, and it certainly is much smaller than GenImage (I recall in million scale). Thus the magnitude of the proposed dataset is not a SOTA point.
> >
> > I also agree with Reviewer L6tg that the dataset should be made available in a solid way.

---

> > > ### Author Response · Authors · 2024-11-25
> > > **Response**
> > >
> > > Dear Reviewer QHiA,
> > >
> > > We sincerely appreciate your reply and valuable advice for our rebuttal. We hope our further response can address your concerns.
> > >
> > > >**Q1. The magnitude of the proposed dataset is not a SOTA point.**
> > >
> > > Sorry for the caused confusion. We will classify it in the following four aspects. We will provide a more detailed explanation in the revised version.
> > >
> > > 1. **Motivation.** Our initial objective is to develop an expanded dataset specifically for evaluating AI-generated image detection.
> > >
> > > 2. **AIGCDetectBenchmark.** It should be noted that the AIGCDetectBenchmark encompasses a compilation of all existing test sets for the task of AI-generated image detection. When compared with any of existing test datasets, our Chameleon dataset offers greater breadth and scope.
> > >
> > > 3. **GenImage.** GenImage is a relatively large dataset; however, it primarily emphasizes the ***training set.*** Its test set, which includes Midjourney, SD V1.4, SD V1.5, ADM, GLIDE, Wukong, and VQDM, is fully incorporated ***within the AIGCDetectBenchmark.*** As illustrated in the ***Tab2 of the main paper***, our Chameleon test set is more extensive than any of the test sets contained within GenImage.
> > >
> > > 4. **Valueable.** The dataset holds significant value. As indicated in ***Tab 5 of the main paper***, all existing detection methods are rendered ineffective. Conversely, as presented in ***Tab 3 and 4***, existing test datasets demonstrate that most detectors perform well, with some achieving near-perfect accuracy.
> > >
> > > Consequently, our Chameleon dataset represents a large-scale extension for AI-generated image detection. In addition to its size, our dataset is characterized by its deceptively realistic images, diversity, and high resolution.
> > >
> > >
> > > >**Q2. The dataset should be made available in a solid way.**
> > >
> > > Thanks for the advice! We provide an anonymous link for the Chameleon dataset, please refer to ***Appendix E*** in the revised version for further details.

---

> > > > ### Comment · Reviewer_QHiA · 2024-11-26
> > > >
> > > > The authors have cleared my concerns.
> > > >
> > > > Please modify the paper to clearly state that the size comparison is only by the test set.
> > > >
> > > > I adjust my rating up to the acceptance side.

---

> > > > > ### Author Response · Authors · 2024-11-26
> > > > > **Thank you for your response and constructive advice**
> > > > >
> > > > > Dear Reviewer QHiA,
> > > > >
> > > > > Thank you for your response and constructive advice, which helps a lot to improve our work! We have included a more detailed explanation of the dataset comparisons in our revised version.

---

### Official Review · Reviewer_L6tg · 2024-11-04

**Soundness:** 3
**Presentation:** 3
**Contribution:** 3
**Rating:** 6
**Confidence:** 4

**Summary:**

This work aims to answer whether the AI-generated image detection problem has been solved. There are two main contributions in this submission, first, they propose a more challenging dataset, called Chameleon, to evaluate the generation ability of existing fake image detectors. Then, they propose a new detector with hybrid detection features, called AIDE (AI-generated Image DEtector with Hybrid Feature), that shows improved detection performance on this new challenging dataset, compared with the existing baseline detector.

**Strengths:**

This work's main contribution is that it has proposed a more challenging dataset with diverse categories, with which 9 AI-generated image detectors cannot handle the detection problem well. In the AIGC detection problem, I think it is important to create such a challenging baseline, particularly given the fact that the existing benchmark seems saturated.

Besides, a new detector with hybrid features (including frequency and semantics) is developed, and it shows improved performance on the AIGCDetectBenchmark, the GenImage benchmark, and Chameleon.

**Weaknesses:**

- This dataset only contains 4 categories, some of which only with hundreds of fake photos, this may not effectively cover the detection of a wide range of photos.

- A recent work also adopts a mixture of expert architecture for fake image detection [1], which seems not be included as a baseline. It is supposed to discuss the major differences to AIDE, and serve as a possibly stronger baseline.

- Regarding JPEG comparisons, only QF=95 and QF=90 are evaluated, more real-world compression such as QF=75, and even more severe but practical case e.g. QF=50 are supposed to be evaluated, too.

[1] Liu, Z., Wang, H., Kang, Y., & Wang, S. (2024). Mixture of Low-rank Experts for Transferable AI-Generated Image Detection. arXiv preprint arXiv:2404.04883.

**Questions:**

- Regarding the dataset construction, this work provides 4 principles in p4, one of which is to cover a wide range of categories. I quite agree with the point. However, this dataset only contains 4 categories, scene, object, animal, and human, and in some categories, there may be only hundreds of photos (e.g. Animal fake images = 313). Then, given the limited number of samples and categories, how can the authors ensure whether the images are representative enough (or cover a wide range of images to be detected)?

- Regarding the dataset construction, have the authors strictly obey to the 4 principles as presented in p4, or the generated images have been carefully selected and validated in 9 baseline methods beforehand in order to create such as challenging dataset?

- Experiments show 9 detectors cannot perform well on this dataset, this is good. However, I am eager to know the underlying reasons for their failure. Also, can the authors conduct a more fine-grained experiment that checks the accuracy of each of the 4 categories (scene, object, animal, and human), along with the overall category accuracy? I feel this might provide more insights.

- Regarding the robustness evaluation of the proposed AIDE, if possible, please additionally evaluate JPEG compression with more realistic quality factors e.g. QF=75, QF=50 etc. Similarly, for Gaussian blur, please evaluate sigma=3 and 4.

**Details Of Ethics Concerns:**

NA.

---

> ### Author Response · Authors · 2024-11-21
>
> We sincerely appreciate your insightful reviews and positive comments. We hope our response can address your concerns.
>
> > **Q1. Given the limited number of samples and categories, how can the authors ensure whether the images are representative enough.**
>
> Thank you for pointing that out. We will include more analysis in the revised version.
>
> 1. Our dataset actually consists of diverse categories from four parent categories, namely, animal, human, object, and scene. For example, the animal category includes more specific subcategories, such as cats, tigers, dogs, and others.
>
> 2. Our dataset was constructed with the goal of simulating real-world scenarios, to focus the images indistinguishable from humans. We agree that the animal category contains fewer images that the other ones, this actually reflects the flaw of modern generative models or human perception systems, i.e., the generated animal images are more easily detected, while the other categories are relatively more difficult.
>
>
> 3. As illustrated in ***Fig 1 of the main paper***, our dataset features a diverse array of categories. Unlike previous test sets that focus on relatively singular scenarios, our dataset emphasizes ***real-world scenarios.***
>
> 4. Although our collection of fake animal images comprises only a few hundred photos, they encompass a broad range, as visualized in ***Fig 1 of the main paper***, and effectively pass the ***“Turing test.”*** Additionally, we assessed some recent methods through a ***fine-grained experiment***, which evaluates accuracy across the four main categories, with results shown in the accompanying tables. We present the ***“fake image Acc / real image Acc”*** for detailed analysis. As demonstrated, the performance of the animal category is comparable to that of other categories. This indicates that our data is sufficient for validating detection methods.
>
>
> ||DIRE|PatchCraft|AIDE|
> |:-:|:-:|:-:|:-:|
> |Animal|2.55/99.77|2.53/96.83|22.81/95.47|
> |Human|1.56/99.86|0.90/97.52|24.48/94.13|
> |Object|3.52/99.49|4.32/94.41|23.71/94.97|
> |Scene|2.58/99.66|0.84/96.76|17.33/95.66|
>
> Therefore, the images are representative enough for validating AI-generated image detectors.
>
> > **Q2. CLIPMoLE[Liu et al., 2024] is supposed to discuss the major differences to AIDE, and serve as a possibly stronger baseline.**
>
> >> **Q2.1 The major differences to AIDE.**
>
> Thanks for pointing this out. Our AIDE  differs from CLIPMoLE in three aspects:
>
> 1. **Motivation.** CLIPMoLE's mixture-of-low-ranks is specifically designed for ***parameter-efficient fine-tuning***. As mentioned in ***L71–75 of the main paper***, AIDE is primarily developed for ***identifying AI-generated images originating from multiple factors***. These cues may intuitively arise from various sources, including low-level textures, pixel statistics, and semantic cues.
>
> 2. **Methodology.** CLIPMoLE adapts only the MLP layers of deeper ViT blocks through a combination of shared and separate LoRAs within an MoE-based structure. As outlined in ***L76–80 of the main paper***, AIDE incorporates a scoring module to capture low-level pixel statistics and harness global semantics with CLIP. The fused features are then utilized for AI-generated image detection.
>
> 3. **Features.** CLIPMoLE is limited to encoding image features. In contrast, AIDE extracts both the underlying noise features and high-level semantic features.
>
>
>
> >> **Q2.2 CLIPMoLE[Liu et al., 2024]  serves as a possibly stronger baseline.**
>
> Thanks for the suggestion. The codes and models have not been released yet, we will incorporate them as a baseline if the codes are available, as promised in their paper.
>
> > **Q3. Evaluate JPEG compression with more realistic quality factors e.g. QF=75, QF=50 and Gaussian blur for sigma=3 and 4 on AIDE.**
>
> Thanks for pointing this out. We have further conducted the robustness of AIDE  on AIGCDetectBenchmark as you suggested.
>
> ||QF=95|QF=90|QF=75|QF=50|
> |:-:|:-:|:-:|:-:|:-:|
> |AIDE|75.54|74.21|70.64|69.60|
>
> ||σ=1.0|σ=2.0|σ=3.0|σ=4.0|
> |:-:|:-:|:-:|:-:|:-:|
> |AIDE|81.88|80.35|80.05|79.86|
>
> The results from the two robustness experiment tables clearly demonstrate that our method exhibits a high degree of robustness.
>
> > **Q4. Regarding the dataset construction, have the authors strictly obey the 4 principles as presented in p4.**
>
> Yes, we strictly adhered to our proposed guidelines when constructing the dataset. We did not use any model to validate the dataset's performance beforehand. Instead, after manual annotation, we directly included the images that humans failed to distinguish their sources, i.e., from camera or generative model.

---

> > ### Author Response · Authors · 2024-11-21
> >
> > > **Q5. More fine-grained experiment that checks the accuracy of each of the 4 categories.**
> >
> > Thank you for your advice. We further conducted a fine-grained experiment on recent detectors, displaying the ***“fake image Acc / real image Acc”*** for detailed analysis.
> >
> > ||DIRE|PatchCraft|AIDE|
> > |:-:|:-:|:-:|:-:|
> > |Animal|2.55/99.77|2.53/96.83|22.81/95.47|
> > |Human|1.56/99.86|0.90/97.52|24.48/94.13|
> > |Object|3.52/99.49|4.32/94.41|23.71/94.97|
> > |Scene|2.58/99.66|0.84/96.76|17.33/95.66|
> >
> > As illustrated in the table above, our Chameleon dataset presents significant challenges across each category.

---

> ### Comment · Reviewer_L6tg · 2024-11-24
> **Availability the Chameleon dataset ? And suggest to revise the paper based on rebuttal response.**
>
> I thank the authors for their rebuttal to help alleviate some of my concerns. A main contribution of this work is to create the Chameleon dataset, however, I didn't find the dataset or a link to it after having checked the paper and its supplementary material.
>
> As a paper with a proposed dataset, I believe it is very important and SUPPOSED to **release the dataset** to the public (should be anonymously at the review phase). Otherwise, the community can't reproduce the results and conduct their experiments on this dataset. I'm looking forward to this proposed dataset and will adjust my rating score depending on its availability.
>
> Besides, I strongly suggest the authors reflect rebuttal responses in the **revised paper**, e.g., including discussions on the MoE detector [1] in related works, adding results on QF=75, QF=50 and sigma=3 and 4 in the main body.

---

> > ### Author Response · Authors · 2024-11-25
> > **Thanks for your reply!**
> >
> > Dear Reviewer L6tg,
> >
> > We sincerely appreciate your reply and insightful advice. We hope our responses can address your concerns.
> >
> > >**Q1. It is very important and SUPPOSED to release the dataset to the public.**
> >
> > Thanks for the advice! We provide an anonymous link for the Chameleon dataset, please refer to ***Appendix E*** in the revised version for further details.
> >
> > >**Q2. I strongly suggest the authors reflect rebuttal responses in the revised paper, e.g., including discussions on the MoE detector in related works adding results on QF=75, QF=50 and sigma=3 and 4 in the main body.**
> >
> > Thanks for your pointing out. We have included a discussion of the MoE detector in the updated version, which can be found at ***L115~117 of the main paper***. Additionally, we have incorporated the results for QF=75, QF=50, and sigma=3 and 4 in ***Tab6 of the main paper***.

---

> ### Comment · Reviewer_L6tg · 2024-11-27
>
> Thank the authors for releasing the dataset and for their efforts in revising the paper. Regarding the dataset, a minor suggestion is to organize it in a more structured way, e.g. adding subfolders containing the four categories in the fake folder (currently, all fake images were put in the same folder), which I believe will help the community do more fine-grained analysis conveniently.
>
> Besides, while I tend to accept this work, I feel a bit concerned regarding the high imbalance of fake images gathered from some categories, e.g. only 313 fake images from the Animal category. I hope the authors could add a discussion in the final version regarding the scalability of the four rules, i.e. discussing the feasibility of increasing the number of certain-category fake images, or explaining in the revised paper the main reason accounting for such high imbalance between categories. Otherwise, it seems quite weird at the moment.

---

> > ### Author Response · Authors · 2024-11-27
> > **Thanks for your recognition of our work**
> >
> > Dear Reviewer L6tg,
> >
> > We sincerely appreciate your recognition of our work and insightful comments. We will organize the latest dataset format in accordance with your feedback. Additionally, we will include a more detailed discussion of the dataset in the revised version.

---

### Official Review · Reviewer_DQu6 · 2024-11-08

**Soundness:** 3
**Presentation:** 2
**Contribution:** 3
**Rating:** 6
**Confidence:** 4

**Summary:**

This paper challenges the assumptions in AI-generated image detection by proposing new training and testing settings. It introduces a benchmark called Chameleon to test human perception and shows significant performance drops in nine evaluated models. The paper also presents AIDE, a detector that combines low-level and high-level features, achieving state-of-the-art results but still offering room for improvement.

**Strengths:**

This paper is relatively well-motivated as AI-generated image detection is a crucial issue. A new dataset named Chameleon is proposed for detecting AI-generated images.  I also find the evaluations thorough. The target issues of the paper are meaningful and worth exploring. The motivation is clear. The paper is easy to follow.

**Weaknesses:**

1.Over the past few years, a multitude of expansive and varied collections of AI-generated imagery have been crafted. Undertaking an in-depth analysis to contrast the newly introduced Chameleon dataset against its peers, considering factors like dataset size, utilized generation techniques, would serve to underscore its unique contributions.

2. It would be good if the authors collect fake images of fake news on the Internet and conduct experiments.

**Questions:**

See Weaknesses.

---

> ### Author Response · Authors · 2024-11-21
>
> We sincerely appreciate constructive suggestions. We hope the following response can address your concerns.
>
> > **Q1. Undertaking an in-depth analysis of Chameleon dataset.**
>
> Thanks for the suggestion. We have included a detailed discussion and analysis in ***L58-63 and L256-261 of the main paper.***
> The unique contribution of our Chameleon lies in the following four aspects:
>
> 1. **Deceptively real.** All AI-generated images within the dataset are manually labeled and have successfully deceived human perception.
> 2. **Diversity.** The Chameleon dataset is derived from ***real-world scenarios***, ensuring a broad and authentic representation.
> 3. **High resolution.** Considering that the majority of images have resolutions surpassing ***720P*** and reaching up to ***4K***, the entire dataset exhibits exceptional clarity. And this high resolution poses increased challenges for the identification of artifacts.
> 4. **Large-Scale.** As illustrated in ***Tab 2 of the main paper***, our current Chameleon dataset stands out as the largest available.
>
> > **Q2. It would be good if the authors collect fake images of fake news on the Internet and conduct experiments.**
>
> Thanks for the suggestion. We did not source fake images from fake news on purpose, and we believe this can be treated as a very meaningful future work. However,
>
> 1. As discussed in ***L206~210 of the main paper***, our Chameleon dataset is derived from a wide range of sources, some of them may indeed be from fake news.
>
> 2. We believe the techniques used for creating fake images in the fake news are also: Photoshop and diffusion models, which has been covered by images in our datasets, thus, the models developed for detecting these fake images should be equally applicable to ones in fake news.
>
> We will clarify these aspects in the revised version.

---

### Official Review · Reviewer_Wrdr · 2024-11-12

**Soundness:** 3
**Presentation:** 3
**Contribution:** 3
**Rating:** 8
**Confidence:** 4

**Summary:**

This paper proposes a new dataset for AI-generated image detection named Chameleon. It also proposes a new model for AI image detection. Comprehensive evaluations are conducted.

**Strengths:**

1.A new datasets and evaluations to show the real status of AI-generated image detection are significant. This work shows that this task is far from solved.
2.The proposed Chameleon dataset is of high-quality and from real-world distributions/sources.
3.The proposed method is straightforward and effective. It combines both high-level semantic features and low-level pixel features.
4.The experiments show effectiveness of the method and the hardness of the new dataset.
5.This paper is easy to follow.

**Weaknesses:**

1.The experiments are mainly about the effectiveness of the proposed detection model. Some validation of the new dataset in its visual quality perspective will make the work stronger.  E.g., by human studies and comparisons between datasets.
2.In Table 6, it is better to also list the accuracy numbers under no perturbations for convenient comparison.
3.More recent related work can be compared or discussed.

**Questions:**

Mentioned in the weakness

---

> ### Author Response · Authors · 2024-11-21
>
> We highly appreciate your recognition of our work and are encouraged by your positive comments. We hope our response can address your concerns.
>
> > **Q1. Some validation of the new dataset in its visual quality perspective will make the work stronger. E.g., by human studies and comparisons between datasets.**
>
> Thank you for the advice. When developing the Chameleon dataset, we devoted significant effort to addressing the issues found in previous datasets, as depicted in ***Fig. 1 of the main paper***. Additionally, we invested substantial manpower in manually annotating existing datasets.
>
> 1. **Visualization.**  ***Fig. 1 in the main paper*** illustrates a comparison between our dataset and other existing datasets, clearly demonstrating that our dataset offers a more realistic portrayal.
>
> 2. **Manual Annotation.**  Chameleon is the first dataset with manual annotations. As written in ***L247–249 of the main paper***, our dataset is manually labeled by having each image evaluated by two annotators. We retain the images only when both annotators mis-classify them as real. Notably, all AI-generated images included in the dataset have successfully tricked human perception.
>
> 3. As per your suggestion, we intend to conduct a randomized selection of 500 images from the existing dataset for evaluation by annotators. This assessment will focus on various aspects, such as the visual quality of images, the depiction of scenes, and the determination of whether the images are authentic or generated by artificial intelligence.
>
> > **Q2. In Table 6, it is better to also list the accuracy numbers under no perturbations for convenient comparison.**
>
> Thanks for the suggestion.  We have incorporated it in the revised manuscript.
>
> ||Original|QF=95|QF=90|σ=1.0|σ=2.0
> |:-:|:-:|:-:|:-:|:-:|:-:|
> |CNNSpot|70.78|64.03|62.26|68.39|67.26|
> |FreDect|64.03|66.95|67.45|65.75|66.48|
> |Fusing|68.38|62.43|61.39|68.09|66.69|
> |GramNet|68.67|65.47|64.94|68.63|68.09|
> |LNP|83.84|53.58|54.09|67.91|66.42|
> |LGrad|75.34|51.55|51.39|71.73|69.12|
> |DIRE-G|68.68|66.49|66.12|64.00|63.09|
> |UnivFD|78.43|74.10|74.02|70.31|68.29|
> |PatchCraft|89.31|72.48|71.41|75.99|74.90|
> |AIDE|**92.77**|**75.54**|**74.21**|**81.88**|**80.35**|
>
>
> > **Q3. More recent related work can be compared or discussed.**
>
> Thank you for highlighting this. We selected another recent approach for comparison, namely, FatFormer [1]. We have incorporated the discussion in the related work in our paper.
>
> We take the publicly available FatFormer model. The table below shows the results of ***FatFormer*** and ***AIDE*** on AIGCDetectBenchmark. As shown in the table below, ***AIDE*** surpasses ***FatFormer*** by a large margin.
>
> ||ProGAN|StyleGAN|BigGAN|CycleGAN|StarGAN|GauGAN|Stylegan2|WFIR|ADM|Glide|Midjourney|SDv1.4|SDv1.5|VQDM|Wukong|DALLE2|Avg|
> |:-:|:-:|:-:|:-:|:-:|:-:|:-:|:-:|:-:|:-:|:-:|:-:|:-:|:-:|:-:|:-:|:-:|:-:|
> |FatFormer|99.89|97.15|99.50|99.32|100.00|99.41|98.80|88.05|78.40|87.98|56.07|67.79|68.03|86.86|73.02|69.70|85.62|
> |AIDE|99.99|99.64|83.95|98.48|99.91|73.25|98.00|94.20|93.45|95.09|77.20|93.00|92.85|95.16|93.55|96.60|**92.77**|
>
> [1] Forgery-aware Adaptive Transformer for Generalizable Synthetic Image Detection. In CVPR, 2024. (arXiv preprint arXiv: 2312.16649)

---

### Meta-Review · Area_Chair_Prjt · 2024-12-21

**Metareview:**

In this work, the authors introduce the Chameleon dataset, a high-quality benchmark designed for AI-generated image detection. They also propose an effective detection method which leverages hybrid features to capture low-level artifacts present in AI-generated images. The authors provide extensive  results demonstrating both the necessity of the dataset and the effectiveness of their detection method.
Although some reviewers raised concerns regarding the novelty and inference speed of the proposed detection method and the comparison between the proposed dataset and existing ones in terms of dataset size, they reach unanimous agreement on the value of introducing a challenging new dataset to drive progress in detection methods under more realistic conditions. Additionally, the paper is received consistently positive ratings, 6.4 on average. Based on a comprehensive evaluation and the significance of the dataset, we have decided to accept the paper. To strengthen the work further, please follow the suggestions from the reviewers.  Reviewer L6tg: Organize the dataset in a more structured manner and discuss how to mitigate dataset imbalance. Reviewer 9Ryx: Clarify the novelty of the detection method in the final paper.  Reviewer QHiA: Clarify the dataset size comparisons is based on the test set.

**Additional Comments On Reviewer Discussion:**

During the discussion period, most reviewers want the authors to compare the proposed method with additional baselines, particularly under more challenging distortion conditions (e.g., JPEG QF=75) to further demonstrate its effectiveness. Additionally, some reviewers raised concerns regarding the novelty of the proposed method and the comparisons between the Chameleon dataset and existing datasets. In response, the authors have provided more results and clarifications, addressing these concerns. This leads to positive ratings above the borderline accept from all reviewers, ultimately raising the average score to 6.4.

---

### Decision · Program_Chairs · 2025-01-22

Accept (Poster)